# Hands-On Gardening in Childcare Centers to Advance Preschool-Age Children’s Fruit and Vegetable Liking in Semi-Arid Climate Zone

**DOI:** 10.3390/ijerph21111485

**Published:** 2024-11-07

**Authors:** Muntazar Monsur, Mohaimen Mansur, Nazia Afrin Trina, Nilda Cosco

**Affiliations:** 1Department of Landscape Architecture (DoLA), Davis College of Agricultural Sciences and Natural Resources, Texas Tech University, 2904 15th St., Lubbock, TX 79409, USA; ntrina@ttu.edu; 2Institute of Statistical Research and Training (ISRT), University of Dhaka, Dhaka 1000, Bangladesh; 3Department of Landscape Architecture and Environmental Planning, College of Design, North Carolina State University, 50 Pullen Road, Raleigh, NC 27695, USA; ngcosco@ncsu.edu

**Keywords:** gardening, childcare, fruit and vegetable (FV) liking, FV knowledge, preschool-age children or preschoolers, obesity, semi-arid climate

## Abstract

Hands-on gardening is linked with healthy eating behaviors, increased outdoor activities, and overall well-being, all contributing factors to preventing obesity. Although these positive associations are widely established for adults and school-aged children, little evidence can be found on how such relationships may extend to early childhood, especially in the preschool years (3–5 years). One recent study conducted in North Carolina (NC) showed that participating in hands-on gardening significantly increased preschoolers’ accurate identification of fruits and vegetables (FV) and FV consumption compared to children who did not participate in hands-on gardening, but no association was found between participation in hands-on gardening and the children’s liking (eating preferences) of FV. FV identification and liking during the early years may lead to lifelong healthy eating behaviors and preferences, making hands-on gardening a critical health intervention in early childhood environments, especially in childcare centers where a majority of the 13 million children in the United States between the ages of 0 and 5 spend most of their waking hours. This research aimed to investigate how hands-on gardening in childcare centers may impact preschoolers’ (3–5 years old) FV knowledge (identification) and liking in a semi-arid climate zone with a high concentration of Hispanic families by conducting experimental research with eight childcare centers and one hundred forty-nine children (*n* = 149) in Lubbock County, located in West Texas. The findings showed changes in average liking scores are generally positive in the experimental group of children who participated in hands-on gardening (compared to the control non-gardening group), implying an improved liking. The findings indicate that the benefits of hands-on gardening in childcare centers for enhancing healthy eating preferences are evident even in a semi-arid climate zone, where high temperatures and limited rainfall present significant gardening challenges. This research underscores the importance of integrating hands-on gardening into childcare programs, highlighting its potential as an effective obesity prevention strategy not only within the US but also in other regions with similar environmental constraints.

## 1. Introduction

This study investigates the effectiveness of childcare hands-on gardening as an obesity-prevention health intervention in a semi-arid climate zone by advancing children’s healthy eating preferences (liking) of various fruit and vegetable (FV) items. Tackling early childhood obesity is critical due to its long-term health implications, with a growing body of evidence suggesting children who are overweight or obese in the early years are highly likely to remain obese in their adulthood [1,2]. Obesity in young children significantly increases the risk of developing chronic conditions such as diabetes, cardiovascular diseases, and mental health issues later in life [3,4]. Childhood obesity predicts long-term risk of adult diabetes [5], and the effect is found to be independent of adult obesity [6]. In the US, the total estimated cost of diabetes alone in 2017 was $327 billion, including $237 billion in direct medical costs and $90 billion in reduced productivity [7], demonstrating the critical importance of preventing early childhood obesity for the country’s health expenditure and economic welfare. Beyond physical health, early obesity is linked to social stigmatization and low self-esteem, which can hinder a child’s emotional development [8,9]. Innovative approaches addressing this issue offer an opportunity to establish healthy habits that can last a lifetime, reducing the likelihood of obesity-related complications in adulthood. Approximately 32% of US children ages 2–19 are overweight or obese, which highlights a major public health problem likely to continue for many years [10]. A simulated growth trajectory based on the 2016 prevalence of childhood obesity predicts that 57% of today’s children will be obese by the age of 35 years [11]. Although obesity prevalence among 2- to 5-year-olds is lower (12.7%) [12] compared to the overall youth (2–19 years) obesity rate in this country, it is still considered a critical stage in human development for combating obesity.

One of the most effective strategies for preventing early childhood obesity is fostering increased physical activity and healthy eating behaviors in childcare settings, two protective/preventive behavioral traits in early childhood significantly associated with reducing obesity risks. With as many as 13 million children [13] spending a significant portion of their time in these environments in the US, childcare settings are crucial venues for early health promotion. Research shows that the quality of early care settings leads to positive outcomes in later life trajectories [14,15]. By incorporating structured physical activity and providing nutritious meals, childcare centers can help combat sedentary behaviors and poor dietary habits contributing to obesity. Additionally, childcare settings can provide early learning on healthy living, instilling knowledge and behaviors that extend into their home environments and communities [16]. These efforts are pivotal because early childhood represents a window of opportunity for shaping lifelong health behaviors. Comprehensive approaches in childcare settings can thus play a significant role in reversing the rising trend of childhood obesity.

The health benefits of hands-on gardening by children while attending childcare centers have gained attention and popularity in recent years, especially for the potential to increase the children’s physical activity and healthy eating behaviors. Gardening is already recognized as an environmental intervention associated with higher levels of healthy eating behaviors for adults [17,18] and school-aged children [19,20]. Although there is limited proof that such associations may extend to early childhood, it is worth exploring how hands-on gardening in childcare centers would perform as a health intervention strategy to advance healthy eating preferences in the early years.

When we think of gardening as an environmental intervention to promote health and well-being, the major concern circles around its potential to change/advance healthy behaviors as a year-round strategy. Figure 1 below shows the latest Plant Hardiness Zone Map for the US published in 2023 by the USDA [21]. Hardiness Zones 6–13 (represented by the colors green, yellow, orange, and red) have an average growing season from 160 days to up to 365 days, showing that for the majority of US childcare settings, gardening can be an effective year-round healthy intervention for preventing the early onset of obesity.

A recent study implementing a Randomized Controlled Trial (RCT) with preschool-age children in North Carolina (NC) showed that the children who received a garden intervention had a greater increase in accurate identification of both fruits and vegetables (FVs), as well as consumption of FVs during tasting sessions, compared to the children who did not receive the garden intervention [23]. The only study (to our knowledge) to test the effectiveness of hands-on gardening as a health intervention in childcare centers in a semi-arid climate zone [24] found that the sedentary behaviors of the children in the “gardening” group were reduced significantly compared to the ”non-gardening” control group of children in West Texas. However, FV knowledge (identification) and liking (eating preferences) of children as an outcome of hands-on gardening have not yet been tested for a semi-arid climate zone. In arid or semi-arid climates, FV gardening can be significantly more challenging compared to a humid subtropical climate zone like NC due to higher temperatures and lower annual rainfall. This is a key reason to conduct this study in West Texas (semi-arid climate) to investigate the generalizability of the findings associated with the positive healthy behavioral outcomes of childcare hands-on gardening in different climate zones in the US.

Not just for its unique climate conditions, the demography of West Texas also makes it a critical area of interest for this study. Texas has the second (after New Mexico) highest concentration (39.1%) of the Hispanic and Latino population. Among 2- to 5-year-old children, Latinos are three times as likely to be obese as Caucasians [25]. Studies also showed that Hispanic children eat less FVs at home even if they and their families show high levels of acculturation [26]. In West Texas, from 2012 to 2022, the Hispanic population grew from 45.8 percent to 53.1 percent [27], making it one of the highest Hispanic concentration areas in the country. How hands-on gardening may alter healthy eating behaviors in the childcare centers located in this area, with its semi-arid climate and Hispanic population, can provide valuable learning for adopting gardening as a health intervention strategy in other areas of the US.

## 2. Materials and Methods

### 2.1. Research Design

This study aimed to assess the effectiveness of a hands-on gardening intervention in licensed childcare centers in promoting preschoolers’ fruit and vegetable (FV) knowledge and preferences in a semi-arid climate zone with a high concentration of Hispanic families. An experimental research design was employed, involving eight childcare centers and 149 children (*n* = 149) in Lubbock County, West Texas. The goal was to corroborate findings from previous studies conducted in NC and to explore whether the positive association between hands-on gardening in childcare and preschoolers’ FV preferences extends to a semi-arid region with a significant Hispanic population. The project utilized a randomized two-group pre- and post-test experimental design [28] as detailed in Table 1.

The core concept of this research design involves the random assignment of subjects (childcare centers) into two groups: an experimental group where participating children would be exposed to hands-on FV gardening and a control group where participating children would not be exposed to any hands-on gardening experience. A total of eight childcare centers were randomly divided, with four centers in each group. Both groups underwent pre- and post-testing to assess children’s FV Knowledge and Liking. However, only the Experimental Group received the intervention, a hands-on FV gardening program in year 1 (2022). The intervention included standardized raised beds (6 beds per center) for selected FV species (built by the research team) and hands-on gardening activities guided by a *Garden Activity Guide* (tailored for the semi-arid climate zone) created by TTU’s partnering organization, the Natural Learning Initiative (NLI) at NC State University. The Control Group centers did not receive any gardening intervention, nor did they receive the guide. During the intervention gardening season, the control centers operated on their regular schedules and experienced no changes in their daily activities. Randomization was intended to ensure that any differences observed in the post-test would be attributed to the experimental variable rather than pre-existing differences between the two groups [28].

This classical experimental design is known for its strong internal validity, allowing the researchers to confidently attribute any changes in outcomes to the intervention itself. By comparing the post-test results, the research team could evaluate the overall effectiveness of the hands-on FV gardening intervention on children’s FV Knowledge and Liking. Additionally, the design enabled an analysis of how both groups changed from pre-test to post-test, helping to determine whether the Experimental Group, Control Group, or neither demonstrated significant differences in FV preference outcomes.

### 2.2. Participating Childcare Sites: Selection and Random Assignment to Groups

The selection criteria for licensed childcare centers in this research included the following: (1) centers must be located in Lubbock County, West Texas, within the semi-arid climate zone; (2) centers must hold a full permit issued by the Texas Health and Human Services (THHS) Child Care Regulation (CCR) during the study period to ensure consistency among participants; (3) centers must enroll preschool-age children (3–5 years old) and have separate classrooms for this age group to facilitate enrollment and data collection; and (4) centers must accept childcare subsidies to ensure participation of children from low-income families and comparability of family income across centers.

A list of 103 licensed childcare centers in Lubbock County was retrieved from the THHS online childcare search tool based on these criteria. The list was then refined to 83 eligible centers based on proximity and enrollment numbers. Driving distance from the university campus was important, as the research team needed to visit centers multiple times for data collection and garden setup. Centers with low enrollment numbers were excluded to ensure comparability in sample size. An online call for applications, facilitated by the Texas Workforce Commission (TWC), was sent to the 83 eligible centers, resulting in 13 completed applications.

From these, eight centers were randomly selected and assigned to two groups: four centers in the Experimental Group, which received the hands-on garden intervention in year 1 (2022), and four centers in the Control Group, which received the intervention a year later in year 2 (2023). This arrangement enabled an experimental model in 2022 to compare pre- and post-intervention FV Knowledge and Liking levels between the two groups while also ensuring that Control Group children had the opportunity to participate in gardening in 2023. Figure 2 illustrates the recruitment process for childcare centers for this study.

### 2.3. Participating Children

In the eight selected childcare centers in Lubbock County, Texas, parents were invited to include their children in the study through the centers’ mail systems. The invitation letter, available both electronically and in print, provided a brief description of the research and outlined its potential benefits. Written consent forms, along with an additional copy for each child, were distributed by classroom teachers to all parents of children aged 3–5 years in the participating centers. Parents who consented to their child’s participation signed and returned the forms at their convenience, placing them in a designated box in the classroom. With the permission of the classroom teacher, the research team collected the signed consent forms from the box. Only children aged 3–5 years were eligible for enrollment in the study. FV Liking data were retrieved from a total of 149 children (76 children from the Experimental Group (E) centers and 73 from the Control Group (C) centers). FV Liking data from 86 children (41 in the Experimental Group and 45 in the Control Group) were deemed unusable for various reasons as described in the limitation section of the paper. In the final analysis, we used comparable data from 63 children (*n* = 63), 35 children from the Experimental Group (E) centers and 28 from the Control Group (C) centers.

The distribution of demographic information of children under study is presented in Table 2. The Experimental Group consisted of an equal distribution of male and female children, but the percentage of male children (64%) was considerably higher in the Control Group. The median pre-gardening intervention age of children in the Experimental Group was almost 9 months higher than the median age in the Control Group. This difference is statistically significant at 1% level of significance. The distribution of ethnicity is also significantly different in the two groups. While Non-Hispanic White or Euro-Americans make up more than half of the children in the Experimental Group, Latino or Hispanic Americans are the majority (64%) in the Control Group.

### 2.4. Variables

#### 2.4.1. Independent Variable: The Garden Intervention

The independent variable in this study is the garden intervention to advance preschool-age children’s FV Knowledge and Liking. The Intervention Garden Components were implemented in selected childcare centers under the Experimental Group (E). This involved the installation of standardized FV garden components with six raised garden beds, with the objective of encouraging children to participate in gardening activities with their teachers and guided by the *Garden Activity Guide*. The physical model of the garden intervention, bed sizes, layout, planting design, etc., followed the model proposed by the NC Study [23]. The six raised planting beds (Figure 3), each measuring 8 feet × 2 feet and 10 inches high, were strategically located outdoors to maximize sunlight exposure and positioned near hose bibs for easy access to water. The planting beds were filled with high-quality, well-drained, and moisture-retentive growing medium to support the cultivation of selected FV. The research team concurrently constructed the FV gardens across all four Experimental Group (E) centers. The selected FVs included six vegetables: cucumbers, green beans, bell peppers, tomatoes, yellow squash, and zucchini, and five fruits: blackberries, blueberries, cantaloupe, strawberries, and watermelon. The selection of FVs depended on several aspects of the research design. First, the selection of FV species was tied to USDA dietary recommendations [29]. In addition, FV species were carefully selected to ensure the possibility of snacking fresh produce from the garden during the hands-on gardening season. That is why the selected FVs are all edible in their raw forms. The NC study [23] also measured FV consumption of children, and year-round availability of the FV was critical. The list of FVs was carefully curated considering children’s dietary needs, suitability for childcare gardening (easy to grow), snackable in raw forms (as garden produce), and availability in the market/grocery year-round. Additionally, as one of the aims of this research was to compare findings with the NC study and to build a comparable study, variables were kept constant as much as possible, including using the same FV species as the NC study [23]. Preschool classroom teachers in the Experimental Group were tasked with leading the *Garden Activity Guide*, designed specifically for children aged 3 to 5 years. This guide encompassed 12 gardening activities grouped into three main themes: Preparation, Maintenance, and Harvesting/Consumption. These hands-on activities involved seed inspection, garden plot preparation, watering, weeding, and harvesting. Teachers facilitated these sessions outdoors, lasting 30 min each, three times a week throughout the primary growing season from late spring to early summer. In contrast, the four Control Group (C) centers did not participate in the gardening activities during the first year but were included in the intervention in the subsequent year. This paper focuses on the data collected in the first year (2022), comparing the Experimental Group (E) and Control Group (C) children before and after the garden intervention to assess the impact of hands-on gardening on advancing FV Knowledge and Liking.

#### 2.4.2. Demographic Variables

Demographic data were collected by research assistants (RAs) at baseline from both the parents and childcare centers. Demographic variables included children’s age, gender, and race/ethnicity.

#### 2.4.3. Dependent Variable: FV Knowledge and FV Liking

To measure FV Knowledge and Liking, we relied on the Fruit, Vegetable Preference Measure tool [30], a non-gendered 5-point face scale designed to depict green peas as a healthy (green) vegetable. The evaluation of these two related variables included showing children 12 different real fruits and vegetables. They were then asked to recognize and name each item, as well as select a face from a non-gendered 5-point face scale (Figure 4) to indicate their liking for each fruit or vegetable. The research assistants recorded responses manually, resulting in the development of two separate measures: FV Knowledge and FV Liking. Pre- and post-intervention “liking” evaluation was based on the same list of FVs (except for apples, which cannot be grown in hands-on gardens in a short period) with the assumption that children were likely to prefer (and eat) more FVs if they participated in hands-on gardening, which may expand their knowledge, exposure, and preferences of a variety of FVs. This assumption is widely supported by research, such as in this systematic review [31] which summarized 10 articles finding statistically significant increases in FV consumption among participants after the implementation of a gardening intervention.

### 2.5. Data Collection Methods

Individual interviews with each child were conducted to assess the dependent variables, FV Knowledge, and FV Liking. A non-gendered 5-point face scale, “super-yummy/super yucky”, was used to assess the FV Liking of children. The research team conducted several preoperational activities before data collection. The TTU research team visited the centers, met with the directors and teachers, and completed collecting all signed consent forms. Once all the consent forms were collected, the research team created separate lists of participating children from each of the eight participating centers. The classroom teacher helped the researcher identify children by name and by introducing them to the researcher. They also convinced each child to participate in the interview and to answer the questions asked by the researcher. Teachers were compensated for each session for their help in the data collection process.

Before data collection, children participated in a storytelling session featuring “Plucky the Pea”, who introduced the rating scale used in the interview process. This narrative was designed to familiarize children with the data collection process and enhance the reliability of their responses.

The interview method (Figure 5) utilized real fruits and vegetables with six fruits (apple, blueberry, blackberry, strawberry, cantaloupe, and watermelon) and six vegetables (cucumber, green bean, red pepper, yellow squash, tomato, and zucchini). The fruits and vegetables were placed on the table individually without adjacent objects to focus the child’s attention solely on the FVs.

Children’s FV Knowledge was measured by showing each FV on the table and asking if they recognized the item (Y/N), followed by a request to name it. Responses were recorded on a digital device (iPad/phone) by the research assistants, including any mispronunciations. For FV Liking, children were asked to indicate their liking using a non-gendered 5-point face scale, showing “super yucky” (1), “yucky” (2), “just ok” (3), “yummy” (4), and “super yummy” (5). This method simplified data entry, reduced errors, and facilitated file conversion for statistical analysis while ensuring strong internal consistency (alpha = 0.79) and acceptable test-retest reliability over a 7–14-day period.

### 2.6. Garden Activity Chart

Garden activity charts (Figure 6) were used to document different garden activities across the four centers. The chart tracked the preschool children’s exposure to fruit and vegetable gardening activities.

Each center received a wall chart measuring 36″ × 48″, organized into cells representing gardening activities (across the top) and calendar dates (along the left), with additional space for weekly notes on the right. The chart was displayed in a location visible to the children. During the morning outdoor garden sessions, teachers reviewed whether any gardening activities had taken place, and if so, stickers were applied to the corresponding cells. Each activity could receive a maximum of five stickers per week, with a total potential weekly score of 60. After five weeks, research assistants (RAs) collected the charts for further analysis.

## 3. Results

### Primary Statistical Analyses

Table 3 reports the change in average number of fruits and vegetables identified by children pre- and post-gardening intervention. Whether it is for fruits or vegetables, the pre-intervention average identification score in the Experimental Group was higher than that in the Control Group. This difference is particularly large in the case of vegetables (1.86 for the Experimental Group to 0.71 for Control Group). All the changes in pre- and post-intervention average scores are positive, implying an increase in average fruit and vegetable identification scores in both groups. Notably, the sizes (magnitudes) of gains are higher in the Control Group in comparison to the Experimental Group (e.g., 2.11 vs. 1.29 in the case of fruits and vegetables).

Changes in the Liking scores have been presented in Table 4. The pre-intervention average Liking scores were slightly higher in the Control Group. However, changes in average Liking scores were generally positive in the Experimental Group (except for fruits), implying improved Liking, but the changes are consistently negative in the Control Group, indicating increased disliking.

The median difference in pre-post gardening intervention scores for fruits and vegetables combined was lower in the Experimental Group than in the Control Group (Figure 7).

The median pre-post intervention difference in Liking score, howevExer, was clearly positive and was much larger in the Experimental Group (Figure 8), suggesting improvement after gardening exposure.

Table 5 contains the results of multiple linear regression for identification of fruits and vegetables. The outcome variable is the post-intervention identification score minus the pre-intervention score, and a positive value indicates improvement in identification following the gardening intervention. After controlling for gender, age, and ethnicity, the average pre-post gardening change in the identification scores was generally lower (by 0.24, 0.47, and 0.71 for fruits, vegetables, and fruits and vegetables combined, respectively) in the Experimental Group in comparison to the Control Group. These changes are, however, not statistically significant at 10% or lower level of significance. The expected post-intervention gain in identification in female children is about 1 fruit or vegetable higher than that of their male counterparts (0.82 and 0.88 with *p* = 0.001 for fruits and vegetables, respectively). Hispanic children were also found to identify significantly more fruits and/or vegetables after intervention when compared to White children.

Table 6 shows that, after controlling for gender, age, and ethnicity, the pre-post change in Liking score for vegetables was two points higher in the Experimental Group in comparison to the Control Group. This increment in pre-post difference is statistically significant at 5% level of significance and indicates a positive effect of the intervention. When both fruits and vegetables were considered together, an improvement in liking that is statistically significant at 10% can also be observed from the gardening intervention.

## 4. Discussion

The findings of this study, as well as its innovative approaches, are significant for many reasons. First, the pairwise comparison shows that after controlling for age, gender, and ethnicity, the Liking score (on a 5-point scale) for vegetables increased by two points in the Experimental Group of children who participated in the hands-on gardening in comparison to the Control (non-gardening) Group of children, and this change was statistically significant at the 5% level (Table 6). This association was also found in the liking of fruits, but it was not statistically significant. When both fruits and vegetables were considered together, the gain of Liking in the Experimental (gardening) Group was statistically significant at the 10% level. This finding indicates that there is a higher chance for preschoolers to like more vegetables if they are exposed to hands-on gardening experiences in their respective childcare centers. A statistically significant association for the liking of just vegetables and not fruits is an unexpected finding and requires further exploration. If we think of what may have mediated this relationship, a seminal study by Matheson, Spranger, and Saxe (2002) [32] explaining preschoolers’ eating preferences and food experiences is insightful. First, preschoolers’ picky eating behaviors are more prevalent when it comes to vegetables rather than fruits [33]. This can be due to the sweet taste of most common fruit species and the children’s familiarity with the taste of them (regardless of their exposure to hands-on gardening). Second, according to Matheson, Spranger, and Saxe (2002) [32], preschool children do not categorize foods based on abstract concepts like nutritional food groups. Instead, their play interactions revealed a tendency to mimic their daily food experiences through realistic and detailed scenarios. This finding suggests that nutrition interventions for preschoolers should be play-based, incorporating familiar, everyday contexts, qualities that can be implanted in childcare hands-on gardening. By embedding gardening interventions within the child’s natural environment (e.g., classroom or home environments), the approach aligns with their developmental understanding and increases the likelihood of preferring and trying more vegetables in their daily lives. Figure 9 below shows how the preschoolers’ food experience may act as a mediator, explaining the most notable finding of this research.

Third, the study compared longitudinal paired data instead of group data to predict behavioral changes regarding FV Knowledge and Liking as an outcome of participation in childcare hands-on gardening. Although the sample size was small to maintain pairwise comparisons, it provided results that can be perceived to have more valid results. Third, the study innovated the Knowledge and Liking survey by using real FVs instead of showing 2D photographs of FVs to children. This innovation is particularly significant for several reasons. This method did not have to rely on preschoolers’ ability to identify FV species from 2D photos taken from just one angle. During the pilot data collection phase of this study, children repeatedly identified a yellow squash as a banana when they were looking at photographs to identify fruits. The idea of using actual FVs (not pictures) for the Knowledge and Liking survey, therefore, provided a greater chance for valid results. Although measuring the validity of the tools was beyond the scope of this research, based on our experience of conducting the survey using real FVs calls for more robust research to increase survey validity for preschoolers by using real FVs instead of photos/pictures.

The NC study [23] demonstrated similar findings that hands-on gardening significantly improved the children’s ability to identify fruits and vegetables. In that study, the children in the intervention group showed greater improvements in FV identification compared to the control group, confirming the positive impact of active participation in gardening activities. However, the NC study [23] differed from this study in the findings of FV Liking. While the NC study observed initial improvements in FV Liking during year 1, these outcomes were not sustained in year 2. In contrast, this study reported a consistent increase in FV Liking among the children in the intervention group, indicating a more lasting impact on their FV preferences. Another major difference between the studies was the inclusion of FV consumption data. The NC study [23] measured FV consumption and found that the children in the intervention group ate more fruits and vegetables than those in the control group. However, this study focused only on FV Knowledge and Liking without measuring actual consumption. Although both studies highlighted the effectiveness of garden interventions in improving FV Knowledge and Liking, they reported the outcomes differently for FV Liking and with/without consumption data.

## 5. Limitations

The challenges and limitations of this study are crucial aspects that highlight the realities of conducting intervention research with preschoolers in childcare centers. One key limitation frequently mentioned is the small sample size, which was influenced by a complex set of factors. These included the logistical constraints of working with multiple childcare centers, the need to obtain parental consent, and the variability in enrollment numbers across centers. Additionally, factors such as geographic location, proximity to the research team for data collection, and capacity of childcare centers all contributed to limiting the number of participants. These constraints underscore the inherent difficulties of achieving large-scale participation in research involving young children in educational settings. Despite these challenges, this study provides valuable insights, but its findings should be interpreted within the context of these limitations. One of the primary challenges we faced was recruiting childcare centers for this study. Despite our efforts, which included building and managing the FV gardens in the Experimental Group (E) centers and offering participation support costs to teachers for each data collection session, few centers fully committed to participation. Childcare centers frequently grapple with high teacher turnover and limited resources. Texas, in particular, has one of the highest turnover rates in early childhood education, exceeding 20% [34]. Even when the childcare center leadership—such as owners or directors—were enthusiastic about implementing hands-on gardening as part of their outdoor environment, they often felt overwhelmed by the “extra effort” required while simultaneously addressing staffing shortages and operational demands. The data collection process following the garden intervention encountered several challenges, particularly due to children leaving or graduating from their childcare centers, which led to significant data loss, especially in the post-intervention phase. Although we were able to recruit 185 children for the study (received signed consent forms from parents), due to various reasons, FV Liking and Knowledge data were initially collected from a total of 149 children. For pairwise comparison, this sample size was further reduced to only 42.3% of the total sample (63 children), limiting our ability to conduct any meaningful analyses at the individual level.

In the final research outcomes, the Experimental Group experienced a notable 37.80% loss of the Knowledge data and 16.91% of the Liking data. Similarly, the Control Group recorded a 21.98% loss of the Knowledge data and an 18.24% loss of the Liking data. A key factor contributing to the overall data loss was the inconsistent response of children for the two dependent variables (FV Knowledge and FV Liking). Specifically, some children provided complete responses for FV Liking using the 5-point face scale but denied responding for FV Knowledge, where they were asked to name or recognize fruits and vegetables. This discrepancy between the two variables further exacerbated the data loss.

Unlike the NC study [23], this research did not measure FV consumption. It is expected that children snacked on some of the items they grew in the gardens during the intervention phase. The “Garden Activity” wall charts maintained by the participating teachers kept records of the children’s snacking from the gardens (please see Figure 6). However, the research team did not have any control over how much (or how often) children ate what they grew in their childcare gardens. Without measuring pre- and post-intervention FV consumption, there was no way for this study to determine whether more liking of FV was associated with more FV eating behaviors in the participating children.

This study was unsuccessful in measuring whether there were any spillover effects of FV liking in the children as an outcome of their exposure to hands-on gardening at their childcare centers. We conducted a parental survey to capture whether the children’s healthy eating preferences changed as a spillover effect from the childcare centers to their home environments or communities. However, the survey was unsuccessful due to the small sample size, with only 22 completed responses across multiple variables. This limited sample size significantly constrained the reliability and generalizability of the findings. Multiple variables in that survey included missing data, and the small number of respondents restricted the statistical accuracy required to identify relevant patterns or trends related to the spillover effects of the intervention.

Last but not least, while this study collected key demographic information like age, sex, and ethnicity, it could not collect data on several other factors, such as the children’s body mass index (BMI) or eating habits (e.g., whether a child is a picky eater or has a good appetite), etc. If the distributions of these unmeasured confounders were different in the Experimental and the Control Groups, this might have impacted the Liking scores of fruits and vegetables in the two groups.

## 6. Conclusions

This study demonstrated that preschoolers in West Texas, a semi-arid climate zone with a high Hispanic population, who participated in hands-on gardening at their childcare centers showed an increased liking for vegetables compared to children who did not engage in gardening activities. These findings suggest that hands-on gardening not only enhances young children’s fruit and vegetable knowledge but also positively influences their food preferences, particularly for vegetables. Given the strong association between vegetable eating preferences, consumption, and reduced obesity risks, this intervention holds promise as a health strategy to combat the early onset of obesity.

To our knowledge, this study is the second one (after the NC study) to investigate the associations between preschoolers’ hands-on gardening and FV knowledge/liking, and it adds to the generalizability of hands-on gardening as a health intervention in childcare centers. The implications extend beyond West Texas, highlighting the potential for hands-on gardening programs to be a valuable tool in early childhood education and obesity prevention on a national scale. As childhood obesity rates continue to rise, especially among minority populations, promoting such interactive and experiential interventions could have a far-reaching impact on the children’s long-term health behaviors. By fostering a connection to healthy foods through direct, playful food experiences, hands-on gardening may become a key strategy for instilling lifelong healthy eating habits in preschool-age children across diverse settings and regions.

## Figures and Tables

**Figure 1 ijerph-21-01485-f001:**
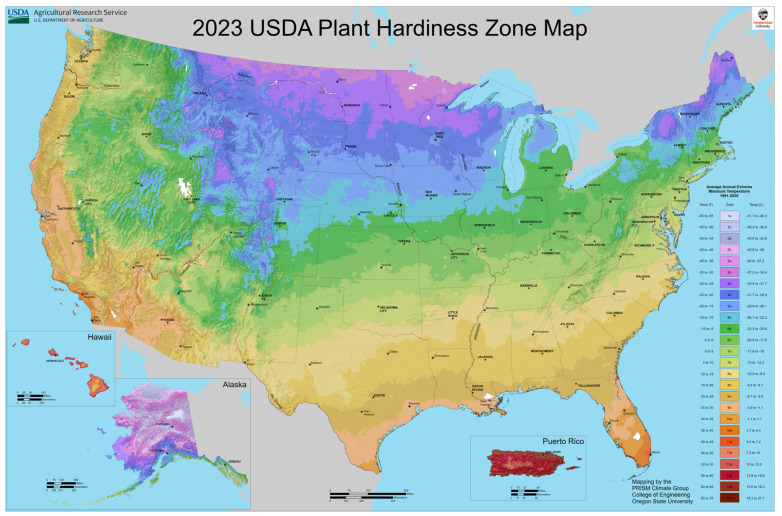
USDA 2023 updated US Plant Hardiness Zone Map [22].

**Figure 2 ijerph-21-01485-f002:**
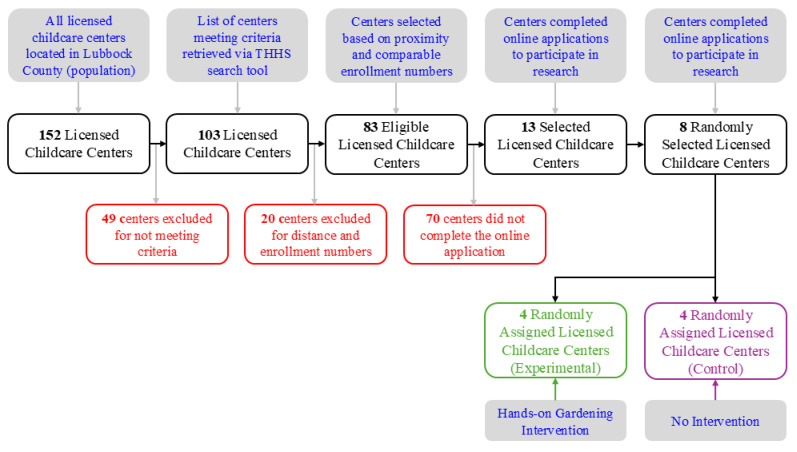
Flow diagram of center selection process.

**Figure 3 ijerph-21-01485-f003:**
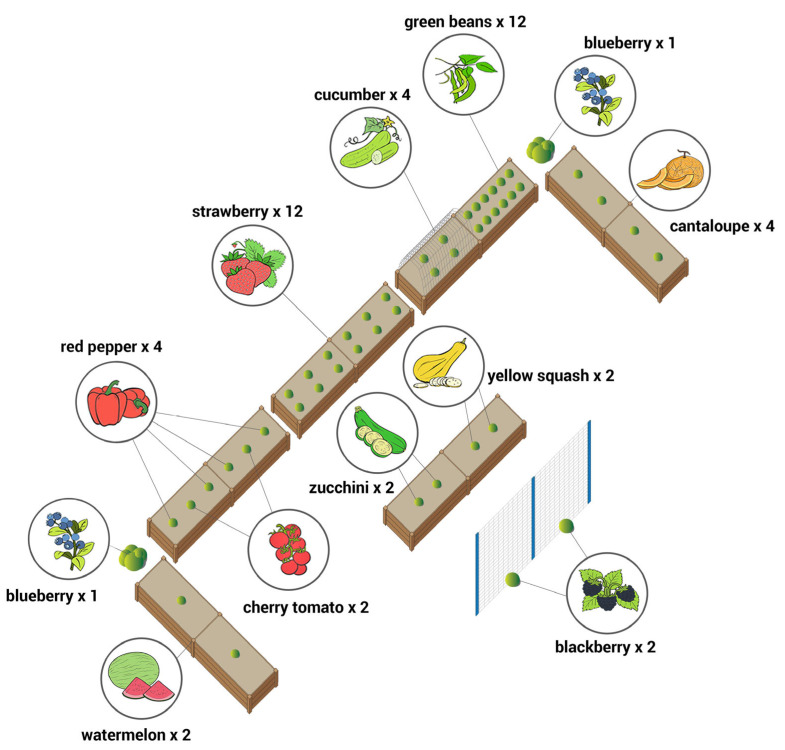
A Standard garden layout (can be changed on-site to fit the available space). The original diagram was created by lead author Dr. Monsur for the NC Study [23].

**Figure 4 ijerph-21-01485-f004:**
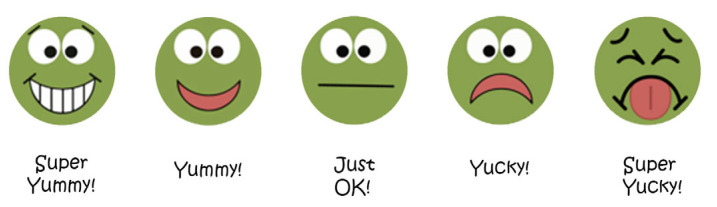
Non-gendered 5-point face scale “super yummy/super yucky”.

**Figure 5 ijerph-21-01485-f005:**
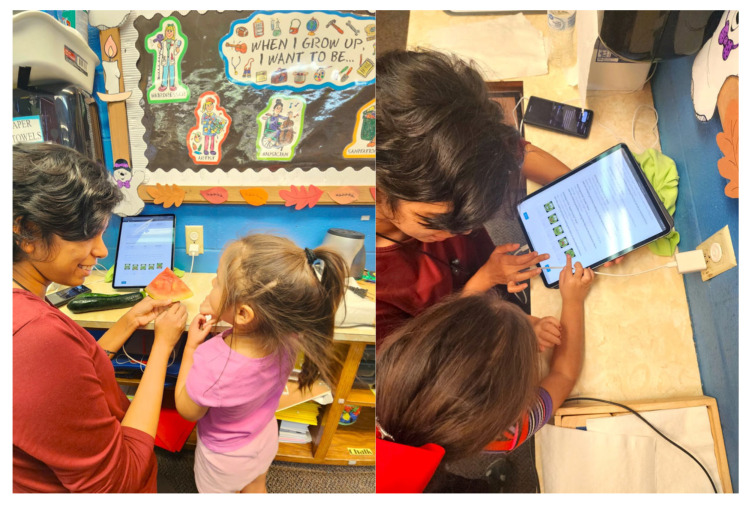
Data collection for FV Knowledge and Liking.

**Figure 6 ijerph-21-01485-f006:**
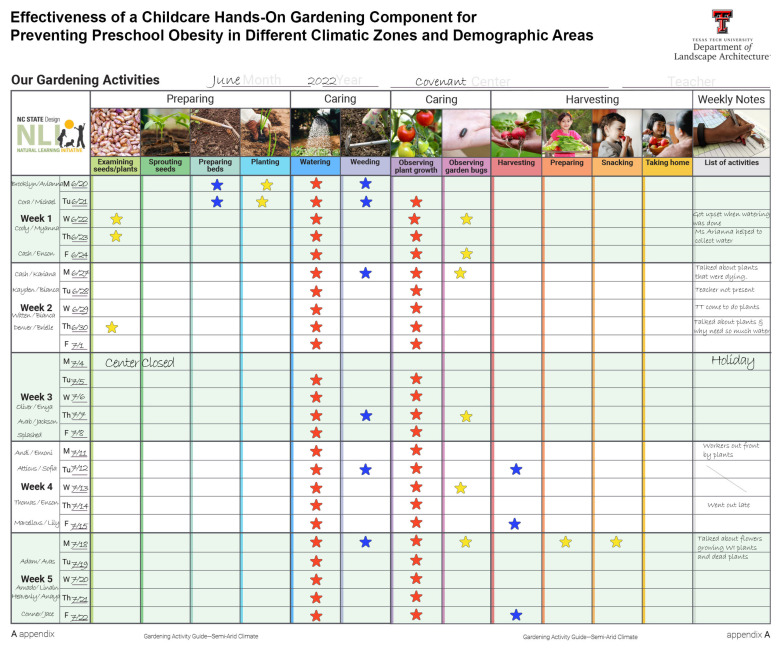
Sample wall chart with recorded gardening activities.

**Figure 7 ijerph-21-01485-f007:**
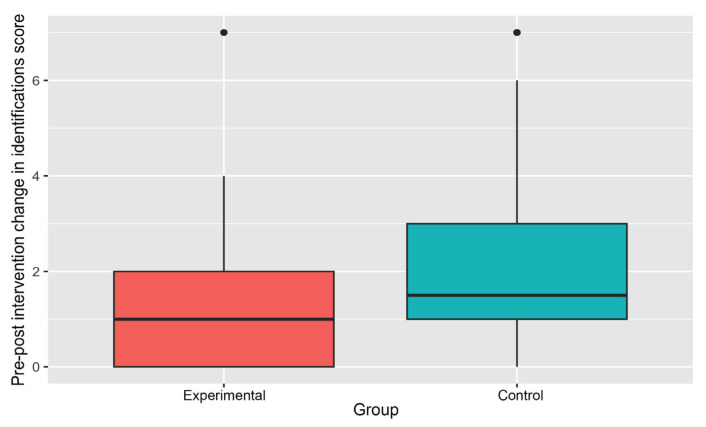
Pre-post intervention changes in identification score (fruits and vegetables).

**Figure 8 ijerph-21-01485-f008:**
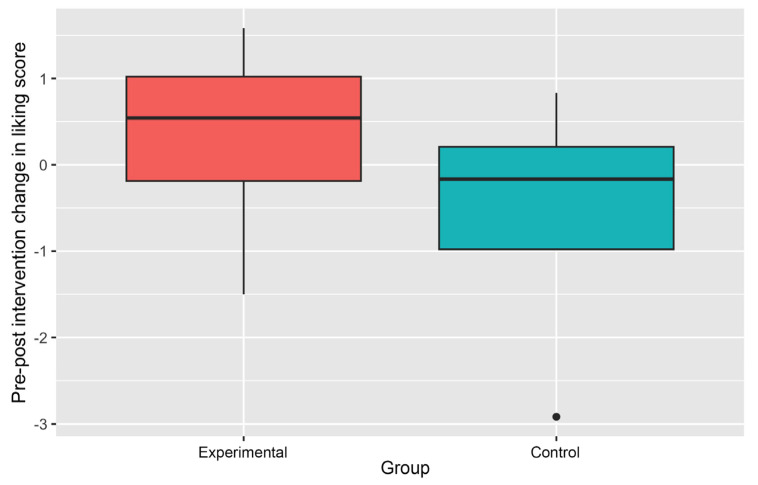
Pre-post intervention changes in liking score (fruits and vegetables).

**Figure 9 ijerph-21-01485-f009:**
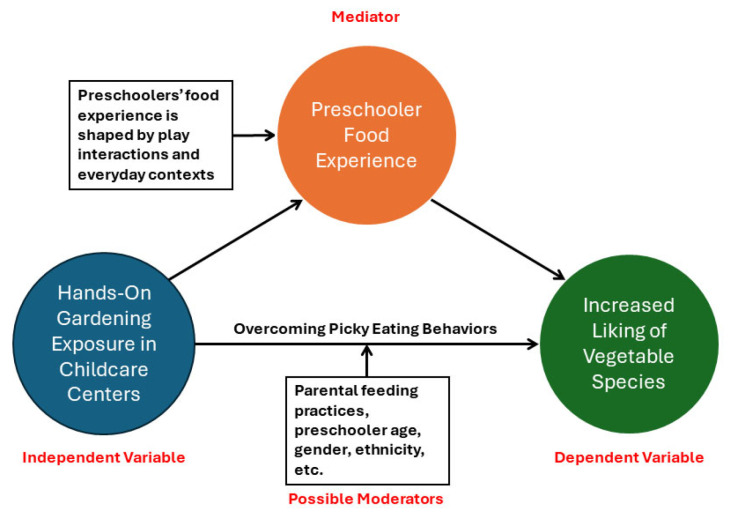
Mediators explaining increased vegetable liking as an outcome of gardening.

**Table 1 ijerph-21-01485-t001:** Two group pre- and post-test experiment.

Group	Pre-Test	Treatment	Post-Test
Experimental Group = E	O_1_	X	O_2_
Control Group = C	O_1_		O_2_

Experimental Group = E = 4 childcare centers with 76 child participants. Control Group = C = 4 childcare centers with 73 child participants. X = Standardized six raised bed FV hands-on garden intervention in summer. O_1_ = Pre-test data (FV Knowledge and Liking data) collection in spring (before garden intervention). O_2_ = Post-test data (FV Knowledge and Liking data) collection in fall (after garden intervention).

**Table 2 ijerph-21-01485-t002:** Distribution of respondent characteristics across Experimental and Control Groups.

Variables	Categories	Experimental Group	Control Group	*p*-Value
Sex	Male	17 (49%)	18 (64%)	0.2
	Female	18 (51%)	10 (36%)	
Pre-intervention age		45.0 (37.0, 49.5)	36.0 (36.0, 41.0)	0.001
Post-intervention age		50 (44, 57)	43 (43, 48)	0.003
Ethnicity	Non-Hispanic White or Euro-American	19 (54%)	7 (25%)	0.01
	Latino or Hispanic American	9 (26%)	18 (64%)	
	Other	7 (20%)	3 (11%)	

Note: The chi-square test of independence is used to test the association between sex and group specification (experimental/control). The association between race and group specification is tested using Fisher’s exact test due to <5 frequency in a cell. Due to a relatively small sample size limitation, possible differences in the distribution of age (pre- and post-intervention) across Experimental and Control Groups is tested using a non-parametric Mann–Whitney U test as opposed to a parametric *t*-test. *p*-values in the last column refer to respective tests.

**Table 3 ijerph-21-01485-t003:** Pre-post mean identification scores and their changes in Experimental and Control Groups.

Items	Average Identification Score		Change
	Pre-Intervention	Post-Intervention	
**Fruits**			
Experimental	3.89	4.43	0.54
Control	3.11	4.04	0.93
**Vegetables**			
Experimental	1.86	2.6	0.74
Control	0.71	1.89	1.18
**Fruits and Vegetables**			
Experimental	5.74	7.03	1.29
Control	3.82	5.93	2.11

Note: The identification score for fruits counts the number of fruits correctly identified out of 6 available fruits and thus ranges from 0–6. With similar definitions, the identification for vegetables ranges from 0–6 and that for fruits and vegetables together ranges from 0–12. The table reports average identification scores per child in the Experimental and Control Groups and their differences in pre- and post-intervention.

**Table 4 ijerph-21-01485-t004:** Pre-post mean liking scores and their changes in Experimental and Control Groups.

Items	Average Liking Score		Change
	Pre-Intervention	Post-Intervention	
**Fruits**			
Experimental	2.39	2.36	−0.03
Control	2.72	2.37	−0.35
**Vegetables**			
Experimental	2.87	3.18	0.31
Control	3.33	2.00	−1.33
**Fruits and Vegetables**			
Experimental	2.48	2.84	0.354
Control	2.85	2.25	−0.604

Note: The liking score is measured on a scale of 1–5. The scores refer to Figure 4 demonstrating the 5-point face scale where a score of 1 indicates “super yucky” and a score of 5 implies “super yummy”. The table reports average liking scores per child in the Experimental and Control Groups and their differences in pre- and post-intervention.

**Table 5 ijerph-21-01485-t005:** Results of pre-post gardening intervention change in identification scores on control variables.

Variables	Categories	Fruits		Vegetables		Fruits and Vegetables	
		Est. Effect	*p*	Est. Effect	*p*	Est. Effect	*p*
Gender (ref: Male)	Female	0.82	0.001	0.88	0.001	1.70	<0.001
Ethnicity (ref: Non-Hispanic White or Euro-American)	Hispanic	0.70	0.01	0.23	0.41	0.94	0.05
	Other	−0.48	0.17	−0.48	0.19	−0.96	0.11
Pre-intervention age (in months)		0.01	0.63	0.01	0.76	0.01	0.64
Group (ref: Control)	Experimental	−0.24	0.37	−0.47	0.11	−0.71	0.13

Note: The outcome/dependent variable in this multiple linear regression is the difference in pre-post intervention identification score. The estimated regression coefficient thus measures the average change in this pre-post intervention difference in score due to a one-unit change in the independent variable (for continuous variables like age) or due to belonging to one group and not the reference group (for categorical variable like sex and ethnicity). *p*-values refer to the *t*-test, where the null hypothesis is that the true effect of an independent variable is zero.

**Table 6 ijerph-21-01485-t006:** Results of regression of pre-post change in Liking scores on control variables.

Variables	Categories	Fruits		Vegetables		Fruits and Vegetables	
		Est. Effect	*p*	Est. Effect	*p*	Est. Effect	*p*
Gender (ref: Male)	Female	−0.63	0.20	0.43	0.51	−0.12	0.80
Ethnicity (ref: Non-Hispanic White or Euro-American)	Hispanic	0.91	0.17	0.59	0.45	0.21	0.73
	Other	0.99	0.27	−0.51	0.61	−0.40	0.60
Pre-intervention age (in months)		−0.04	0.26	0.02	0.72	0.03	0.49
Group (ref: Control)	Experiment	0.92	0.19	2.05	0.04	1.20	0.10

Note: The outcome/dependent variable in this multiple linear regression is the difference in pre-post intervention Liking score. The estimated regression coefficient thus measures the average change in this pre- and post-intervention difference in score due to a one-unit change in the independent variable (for continuous variables like age) or due to belonging to one group and not the reference group (for categorical variable like sex and ethnicity). *p*-values refer to the *t*-test, where the null hypothesis is that the true effect of an independent variable is zero.

## Data Availability

The data presented in this study are available upon request from the corresponding author (M.Mo.).

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
