# Peer review of "Hands-On Gardening in Childcare Centers to Advance Preschool-Age Children’s Fruit and Vegetable Liking in Semi-Arid Climate Zone"

_ijerph, 2024, doi:10.3390/ijerph21111485_

Round 1
Reviewer 1 Report
Comments and Suggestions for Authors
This study aimed to evaluate the impact of a hands-on gardening program in licensed childcare centers on enhancing preschoolers' knowledge and preferences regarding fruits and vegetables (FV) in a semi-arid region with a majority of Hispanic population. A notable strength of this research is the thorough description of the Hands-on Garden Intervention in the methodology section, along with the successful outcomes, which underscore the potential of such programs as a valuable tool for early childhood education and obesity prevention both nationally and globally. However, in my opinion the paper has some shortcomings in regards to some methodology and data analysis. Specific comments follow.
1. You should mention your statistical method either at the end of the methods section or in the footnote of each table.
2. Line 140-141: control group where participating children will not be exposed to any hands-on gardening experience. More details about the control group need to be provided. For example, when the experimental group receiving gardening in intervention, what did control group receive? Nothing at all? Were they provided with any nutrition classes or dietary recommendations during mealtime?
3. Line 247-248: Demographic variables included children's age, gender, race/ethnicity, and parental education. On Table 6, Why didn’t include parental education as a control variable?
4. Inter-personal variances between two group? For example, children’s bodyweight status, how many picky eater in each group? Which also were confounders for this study. Need discuss more in your discussion part.
5. Line 253: I’m very like the non-gendered 5-point face scale (Figure 4) in this study. Just curious, why you choice green color is instead of other colors? Yellow?
6. Compared to NC’s study, given that both use the same F/V but are conducted in different climate zones. I believe that using more locally grown fruits and vegetables suited to a semi-arid climate would be more effective.
Author Response
Please review the attached Word file.

Reviewer 2 Report
Comments and Suggestions for Authors
Dear Authors,
I think it is an interesting idea. The article is well organized and well explained. Probably, the authors of the article, in addition to knowing the science of architecture and agriculture, also had the skill of agriculture in their childhood. In general, children's contact with soil and plants, besides meeting the goals of the authors of the article, increases their body's resistance to allergens and reduces their respiratory diseases in adulthood. Considering that the target is children, it is necessary to add some information in the article regarding the observance of health principles and measures, such as checking vaccination against tetanus, having work clothes, physical examination to avoid open wounds on hands and feet, and periodic parasite testing. In countries that have been important to agriculture for a long time, such as Iran, in ancient times, such measures were used to gain positive energy in children. I think you should also implement it for children with autism spectrum. According to Avicenna, a renowned Iranian physician and philosopher, "When you do not know the nature of the malady, leave it to nature; do not strive to hasten matters. For either nature will bring about the cure or it will reveal itself clearly what the malady really is (ca. 970–1037)"
I suggest the following reference to be used in the article and to conduct such research:
Nedaeinia, R. et al. (2022). Lifestyle Genomic interactions in Health and Disease. In: Kelishadi, R. (eds) Healthy Lifestyle. Integrated Science, vol 3. Springer, Cham. https://doi.org/10.1007/978-3-030-85357-0_3
Author Response
Please review the attached Word file.

Reviewer 3 Report
Comments and Suggestions for Authors
see attached file.

Round 2
Reviewer 1 Report
Comments and Suggestions for Authors
I am very satisfied with the author's reply. No further question!